# Behaviour change physiotherapy intervention to increase physical activity following hip and knee replacement (PEP-TALK): study protocol for a pragmatic randomised controlled trial

Toby O Smith [1,2] Scott Parsons,[1] Beth Fordham,[1] Alexander Ooms,[1,3] Susan Dutton,[1,3,4] Caroline Hing,[5] Vicki S Barber,[1,3] May Ee Png,[1,3] Sarah Lamb,[6] on behalf of the PEP-TALK Trial Collaborators

For numbered affiliations see end of article.

**Correspondence to**
Dr Toby O Smith;
toby.smith@ndorms.ox.ac.uk

## ABSTRACT

**Introduction** While total hip replacement (THR) and total knee replacement (TKR) successfully reduce pain associated with chronic joint pathology, this infrequently translates into increased physical activity. This is a challenge given that over 50% of individuals who undergo these operations are physically inactive and have medical comorbidities such as hypertension, heart disease, diabetes and depression. The impact of these diseases can be reduced with physical activity. This trial aims to investigate the effectiveness of a behaviour change physiotherapy intervention to increase physical activity compared with usual rehabilitation after THR or TKR.

**Methods and analysis** The PEP-TALK trial is a multicentre, open-labelled, pragmatic randomised controlled trial. 260 adults who are scheduled to undergo a primary unilateral THR or TKR and are moderately inactive or inactive, with comorbidities, will be recruited across eight sites in England. They will be randomised post-surgery, prior to hospital discharge, to either six, 30 min weekly group-based exercise sessions (control), or the same six weekly, group-based, exercise sessions each preceded by a 30 min cognitive behaviour approach discussion group. Participants will be followed-up to 12 months by postal questionnaire. The primary outcome is the University of California, Los Angeles (UCLA) Physical Activity Score at 12 months. Secondary outcomes include: physical function, disability, health-related quality of life, kinesiophobia, perceived pain, self-efficacy and health resource utilisation.

**Ethics and dissemination** Research ethics committee approval was granted by the NRES Committee South Central (Oxford B - 18/SC/0423). Dissemination of results will be through peer-reviewed, scientific journals and conference presentations.

**Trial registration number** ISRCTN29770908.

## INTRODUCTION

Total hip replacement (THR) and total knee replacement (TKR) are the two highly successful orthopaedic procedures which

---

### Strengths and limitations of this study

► The effectiveness of a behaviour change physiotherapy intervention to increase physical activity compared with usual rehabilitation after total hip replacement or total knee replacement will be demonstrated with a pragmatic clinical trial design.

► Functional, behavioural and psychological outcomes will provide evidence to determine the mechanisms by which the intervention is or is not effective.

► A multicentre recruitment approach will provide greater external validity across population characteristics in England.

► It is not possible to blind participants to the rehabilitation treatments given the participatory nature of the interventions.

► The group-based intervention may be challenging to ensure sufficient numbers within each group as participants enter the trial.

---

reduce pain for people with osteoarthritis.[1 2] Over 206 000 THR and TKRs were performed in UK in 2018.[1] Approximately 90% of patients are satisfied following THR and TKR,[2] with significant improvements in pain and physical function after 3 to 12 months.[2 3]

Historically, it has been assumed that people become more active following THR or TKR through the amelioration of joint pain.[4] However, the current literature suggests physical activity, at best, remains the same from preoperatively to postoperatively and in some instances declines.[4 5]

People following THR and TKR have reported a number of challenges which make them engaging in physical activity difficult, most notably psychosocial barriers and fear of avoidance beliefs.[6] Such barriers include receiving insufficient and inconsistent

information on being more physically active, fear of damaging joint replacements and causing pain and not being able to goal-set or problem-solve physical activities within individual's lifestyles.[6] While previous international guidance has acknowledged the importance of physical activity on health and well-being, people following THR and TKR have acknowledged these difficulties in being more active.[6] They have cited limited support or guidance currently offered on how to overcome these problems postoperatively.[6]

Not being physically active after joint replacement can have a major negative impact on a person's health and a burden on the National Health Service (NHS). Medical comorbidities are common in this population. These include hypertension (56%),[7] cardiovascular disease (20%),[8] diabetes (16%)[8] and multi-joint pain (57%).[7] Twenty-seven per cent of people who undergo joint replacement have three to four comorbidities.[8] Medical comorbidities have a significant negative impact on both health-related quality of life (HRQoL) and result in a societal burden.[9 10] Participating in regular physical activity can decrease the risk of cardiovascular disease by 52%,[11] diabetes by 65%[12] and some cancers by 40%.[13] It is associated with a reduction in all-cause mortality by 33% and cardiovascular mortality by 35%.[14]

Current rehabilitation following THR and TKR in the UK, as advocated by the British Orthopaedic Association, centres around regaining joint movement, strength and gait (walking pattern) re-education.[15] There is currently no evidence informing patients or healthcare professionals on how to increase physical activity following THR and TKR. Previous research has demonstrated that behaviour-change interventions can effectively increase physical activity across the lifespan.[16–18] However, following joint replacement, people have specific psychological needs and challenges which differ to the non-joint replacement population.[6] Therefore, a specific intervention tailored to this population's health beliefs, including fear avoidance regarding implant survival, dislocation and increased knowledge on the impact of physical inactivity on other comorbidities, is required. Accordingly, the purpose of this trial is to answer the research question 'following a primary THR or TKR, does a group exercise and behaviour-change intervention targeted to increase physical activity participation increase HRQoL and clinical outcomes over the initial 12 postoperative month compared with group exercise alone?'

## METHODS AND ANALYSIS
### Trial design
This is a two-arm, pragmatic, parallel, multicentre, randomised controlled superiority trial (RCT) to assess the effectiveness of a group exercise and behaviour-change intervention aimed to increase physical activity in people following THR or TKR. Nine UK NHS hospitals are involved in the trial across England. The study flow

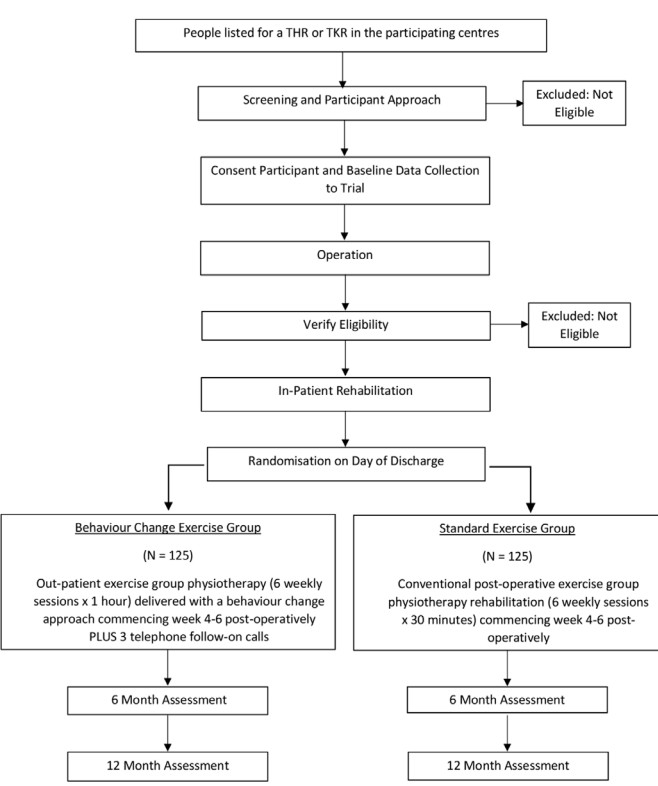

**Figure 1** Study flow chart. THR, total hip replacement; TKR,total knee replacement.

chart is presented as figure 1. Table 1 presents a summary of trial objectives, outcome measures and time points.

### Trial participants
A minimum of 260 participants will be recruited.
Participants are eligible if they are:
► Due to undergo primary (first time) unilateral THR or TKR where the indication for surgery is degenerative joint pathology (not trauma).
► Aged 18 years and over.
► Classified as 'moderately inactive' or 'inactive' using the General Practice Physical Activity Questionnaire (GPPAQ).[19]
► Have a Charlson Comorbidity Index (CCI) of ≥1 point.[20 21]
Participants are ineligible if they have:
► An absolute contraindication to exercise such as severe cardiovascular or pulmonary disease.
► Cognitive impairment defined as an Abbreviated Mental Test Score (AMTS)[22] of <8.
► A usual place of residence that is a care home.
► Already enrolled onto another trial investigating physical activity, exercise adherence or behavioural therapy interventions.
► An inability to read and/or comprehend English.
► No access to a working telephone.

### Recruitment
Potential participants will be identified from UK NHS hospital trusts by the clinical team once they have been listed for THR or TKR. They will be asked whether they

**Table 1** PEP-TALK trial objectives, outcome measures and measurement time points

| Objectives | Outcome measures | Time points |
|---|---|---|
| **Primary objective** | | |
| To compare physical activity participation at 12 months post randomisation between people who receive a group exercise and behaviour change intervention versus group exercise alone following THR or TKR. | University of California, Los Angeles Activity Scale | Baseline; 6 months; 12 months (primary time point) |
| **Secondary objectives** | | |
| To compare the functional outcome between people who receive a group exercise and behaviour-change intervention versus group exercise alone following THR or TKR. | Lower Extremity Functional Scale | Baseline; 6 months; 12 months |
| To compare the disease-specific pain and function between people who receive a group exercise and behaviour-change intervention versus group exercise alone following THR or TKR. | Oxford Hip Score[36] or Oxford Knee Score | Baseline; 6 months; 12 months |
| To compare the perceived level of pain between people who receive a group exercise and behaviour-change intervention versus group exercise alone following THR or TKR. | Numerical Rating Scale for Pain | Baseline; 6 months; 12 months |
| To compare participant's self-efficacy between those who receive a group exercise and behaviour-change intervention versus group exercise alone following THR or TKR. | Generalised Self-Efficacy Scale | Baseline; 6 months; 12 months |
| To compare participant's fear avoidance to movement between those who receive a group exercise and behaviour-change intervention versus group exercise alone following THR or TKR. | The Tampa Scale for Kinesiophobia | Baseline; 6 months; 12 months |
| To compare participant's psychological distress (anxiety and depression) between those who receive a group exercise and behaviour-change intervention versus group exercise alone following THR or TKR. | Hospital Anxiety and Depression Scale | Baseline; 6 months; 12 months |
| To compare participant's complications and adverse events between those who receive a group exercise and behaviour-change intervention versus group exercise alone following THR or TKR. | Complications or adverse events recorded in the case report forms | 6 months; 12 months |
| To collect cost-effectiveness data (health resource utilisation; direct and indirect costs) of a group exercise and a behaviour-change intervention versus group exercise alone following THR or TKR. | EQ-5D-5L; Bespoke healthcare resource self-reported questionnaire | 6 months; 12 months |

would like to know more about the PEP-TALK trial. There are two options for this: either they will be given a copy of the Participant Information Sheet (PIS) and asked to contact a member of the Clinical Research Network (CRN) research team who will provide more information or will be asked whether they are happy for a member of the CRN research team to contact them directly with more information about the trial. Verbal consent for the approach will be documented by a clinical team member in the medical notes.

Potential participants will be asked to read the PIS and asked to discuss their potential participation with anyone who they feel would provide useful advice such as friends, family member or carers. The number of people provided with the PIS will be recorded to monitor how many participants are assessed for initial eligibility and sent the PIS.

Eligible patients who agree to participate will provide their consent during the preoperative assessment appointment. Participant's eligibility will then be verified by reviewing the medical notes and by interviewing

the participant using the screening log, CCI and GPPAQ tools.

Written informed consent (online supplementary file 1) will be obtained prior to any trial-specific procedures being performed. The baseline case report form (CRF) will be completed at the preoperative assessments once consent has been taken. After the TKR or THR, eligibility will be confirmed by the site research team member reviewing the postoperative notes.

### Randomisation, blinding and allocation concealment

Consented participants will be randomised (1:1) to the group exercise and behaviour-change intervention (intervention group) or group exercise alone intervention (control group) using the centralised computer randomisation service RRAMP (https://rramp.octru.ox.ac.uk) provided by the Oxford Clinical Trials Research Unit (OCTRU). This will either be undertaken directly by the site's research facilitator or by contacting the trial office over the telephone, who will access the system on their behalf. Randomisation will be undertaken using a minimisation algorithm to ensure balanced allocation of participants across the two treatment groups, stratified by:

► Hospital site.
► Type of joint replacement (THR or TKR).
► CCI of 1 to 3 versus ≥4.[20 21]

The minimisation algorithm will be seeded using simple randomisation and will have a probabilistic element introduced to ensure unpredictability of treatment assignment.

Baseline data will be collected during the preoperative assessment appointment and prior to randomisation. Therefore, the central randomisation system will issue a screening identifier (ID). Once the participant has been randomised, the central randomisation system will provide a participant ID to be used on all subsequent data collection forms. Sites will be responsible for linking the screening ID to the participant ID, with all linking information remaining at sites. This will be double-checked across the CRFs to ensure that the correct paperwork has been provided to the correct participant.

Due to the nature of the intervention, participants and those delivering the rehabilitation will be aware of the treatment allocation. By virtue of this design, it is not possible to blind participants, physiotherapists or the site researchers.

*Change to the randomisation allocation*: An amendment was made to the randomisation ratio in the minimisation algorithm from 1:1 to 2:1 (Experimental Intervention:Usual Care) after 75 participants had been randomised. The expected allocation ratio at the end of the trial is likely to be approximately 1.5:1. This was done to ensure that a greater number of people are allocated to the experimental intervention, which is a group-based intervention. As the experimental intervention was designed to have three or more people per group, early sites have found it difficult to consistently reach this level of participant numbers with the original 1:1 randomisation

allocation. This change was made to reduce the risk of small numbers in the experimental group. The Trial Management Group (TMG) agreed this on 15th August 2019. The sample size was increased to 260 to account for this change. This was approved by the sponsor, Data and Safety Monitoring Committee (DSMC) and research ethics committee.

### Intervention

#### Usual care

This will be received by both control and intervention groups.

In the inpatient setting, all participants will be seen by a physiotherapist a minimum of daily on weekdays and will follow their hospital's standard postoperative pathway. Average hospital length of stay for people after THR and TKR is 5 days.[1] Rehabilitation consists of gait re-education, exercises and advice regarding transferring from bed to chair, toileting and dressing. Advice on continuation of gait progression and activities of daily living are encouraged by the physiotherapy team. Once considered medically fit and safe by the multidisciplinary team, patients are discharged. These follow current, routine, nationwide practice.[23]

All participants attend six weekly, 30 min group-based exercise classes within each hospital trust's physiotherapy department after THR or TKR. These groups consist of up to 12 people and commence within 4 weeks post-randomisation. The principles regarding prescription of group exercises to increase range of motion, strength and gait pattern are consistent. While the rehabilitation of THR and TKR focusses on overall lower limb function, all participants following a THR focus on hip exercises, whereas those following a TKR focus on knee exercises. For this trial, exercise diaries will be provided and maintained by participants to monitor exercise performance and compliance within and outside the exercise groups for the 6-week intervention period.

#### Experimental intervention

Participants randomised to the experimental group will receive the same in-hospital care and physiotherapy prior to hospital discharge. They will also receive the same six weekly, group-based 30 min exercise session. The only difference between the two groups is the addition of a 30 min group-based (up to 12 people) behaviour change approach intervention prior to the routine 30 min of exercise and three telephone follow-up calls 2, 4 and 6 weeks after the last group-based session (the PEP-TALK intervention).

The PEP-TALK intervention is theoretically-based within the cognitive-behavioural model of understanding. It uses evidence-based behaviour change techniques to target internal (cognitions and behaviours regarding physical activity) and external factors (social and environmental barriers). It specifically aims to target self-efficacy beliefs and fear-avoidant behaviours. A senior health psychologist (BF), physiotherapist/cognitive-behavioural

therapist (ZH) and senior clinical academic physiotherapist (TS) developed an in-person, 1 day training session. The training was delivered by BF and TS to all physiotherapists in the trial who were delegated to deliver the PEP-TALK intervention. The training included role-play, knowledge and understanding testing and supplementary materials to support on-going learning. All physiotherapists were invited to contact the training team (BF, ZH and TS) to clarify any training queries throughout the trial.

In the PEP-TALK sessions, through group discussion, participants and physiotherapists will be encouraged to develop a positive therapeutic alliance, where the physiotherapist will generate an environment of trust and belief around individual challenges participant's have, to support them to overcome these for sustained physical activity adoption. A treatment log will be completed by the physiotherapists to record the components of what is discussed across the group in each session.

The PEP-TALK intervention also includes a home-practice element. Participants will be supported with skills developed in the group, to work at home on challenges, barriers and facilitators to physical activity behaviour. The 'home-work' workbook aims to translate and develop the skills learnt within the group session into daily life activities. The workbook is designed to be used during the six sessions although participants are encouraged to keep this as a record and prompt for long-term behaviour change. The workbook will be referred to in the telephone follow-up calls. The workbook encourages reflective activities such as recording physical, emotional and cognitive barriers and facilitators to physical activity. It also offers problem-solving techniques such as "can you break down a large physical activity (eg, 'getting the house all cleaned for the family coming over') into tasks which you can 'prioritise', 'plan', 'tolerance level' and 'evaluate.'" The workbook also includes other activities to encourage pacing and behaviour modification, goal-setting to the individual's health and social needs and techniques to challenge fear-avoidant behaviours. Education on exercise and the detrimental effects of physical inactivity will also be discussed. Participants will complete this with their home exercise plan which solely focusses on lower limb exercises rather than their behaviours and thoughts around physical activity.

The same trained physiotherapists who deliver the PEP-TALK intervention will, once the participant has completed their group sessions, deliver three, 20 min telephone follow-up calls. These will occur at 2, 4 and 6 weeks after completing the group sessions. During these follow-up telephone calls, participant's goals will be reviewed, any barriers to the completion of these goals will be identified and the physiotherapist will review any 'unhelpful' and 'helpful' thoughts or feelings towards physical activity which may have arisen since the last consultation. Each telephone call will close with the development of longer-term physical activity goal-setting and promotion of empowerment towards physical activity

participation using the behavioural principles instilled during the group intervention. A log will be collected by the physiotherapist to record the components of what is discussed during the telephone calls.

### Delivery

Both the PEP-TALK intervention and exercise groups will be delivered as 'rolling' programmes. This means that new participants can join the group as it runs rather than waiting for a new 'block' of sessions to start. This will allow greater flexibility in allocating participants to join group sessions, while also avoiding a delay between postoperative referral and starting the sessions. The flexible nature of the experimental intervention also makes this feasible, with no ordering of the content of intervention sessions which would preclude such a rolling programme.

### Contamination

The physiotherapists who deliver the behaviour change sessions will be taught the skills required to deliver the experimental intervention. These physiotherapists will not be permitted to deliver the control exercise group intervention during the trial period (and vice versa). This approach mitigates the risk of contamination. Due to the interventions being delivered in an outpatient setting, there is a reduced risk of participants sharing their knowledge and experience of the interventions between the control and intervention groups, further minimising the risk of between-group contamination. Participants in the control group will not be permitted to join the experimental intervention group's exercise sessions (and vice versa). This will also reduce the risk of between-group contamination.

### Co-interventions

During the course of follow-up, participants may require further interventions as part of their recovery following surgery as per routine NHS practice. Further clinical interventions will be permitted for trial participants without the participant having to withdraw. If a participant receives additional treatment to the trial intervention, the details of the treatment received and the reasons will be collected.

### Quality assessment

The trial will be monitored and audited in accordance with the current approved protocol (V. 4.0, 17 September 2019), good clinical practice,[24] relevant regulations and standard operating procedures (SOPs).

All designated physiotherapists who deliver the usual care group exercises will be taught about the standardised control intervention procedures, that is, clarification on the use of exercise diaries and treatment logs which are additional to usual care.

Designated physiotherapists delivering the behaviour change intervention will attend a 1 day, face-to-face course where they will be taught the intervention and processes involved by a member of the PEP-TALK team who developed the intervention (BF, ZH or TS). In

addition, to assess the fidelity to the trial intervention, a health psychologist (BF) will undertake a debriefing telephone call with each physiotherapist after they have delivered their first group session. This will allow the trainer to re-enforce any learning required, address any uncertainties and to identify and correct any variation in the treatment protocol. Finally, each physiotherapist who delivers either the PEP-TALK or control group intervention will be monitored during a site visit at their third or fourth intervention session. The PEP-TALK intervention physiotherapist will be monitored by a health psychologist (BF) or physiotherapist with expertise in health psychology (ZH). Control group interventions will be monitored by a practicing physiotherapist (TS). Sessions will be monitored against the protocol to determine whether there are issues around fidelity, contamination across groups or adherence/compliance of participants. Where further training is required, this will be instigated following these visits. Further monitoring visits will be coordinated as required.

## Assessments
### Baseline assessment
Baseline data will be collected prior to randomisation during the preoperative assessment appointment, once consent has been obtained. This will include: gender, age, measured height and weight, CCI, self-reported presence and location of multi-site joint pain, comorbidities determined from the medical notes, AMTS, employment status and occupation (when appropriate). Participants will also complete measures on: physical activity (University of California, Los Angeles (UCLA) Activity Scale)[25]; physical function (Lower Extremity Functional Scale (LEFS)[26]; disease-specific function (Oxford Hip Score[27] or Oxford Knee Score[28]); pain (numerical rating scale for pain); self-efficacy (Generalised Self-Efficacy Scale[29]); fear of movement or kinesiophobia (Tampa Scale for Kinesiophobia[30]); psychological distress (Hospital Anxiety and Depression Scale (HADS)[31]; HRQoL (EQ-5D-5L[32]) and the health utilisation questionnaire.

Follow-up data will be collected 6 and 12 months post-randomisation. Questionnaires will be sent to participants by post from the trial office and returned using a pre-addressed, prepaid envelope. If participants have not responded within 14 days of posting, the trial team will attempt to telephone the participant on up to two occasions to remind them to complete the questionnaires. If required, a second postage of the questionnaires will be provided if requested by the participant during these follow-up telephone calls. If participants wish not to complete the questionnaires, they will be provided with the opportunity to complete the UCLA Activity Scale and EQ-5D-5L questionnaires over the telephone. If these methods fail, the participant would be categorised as a non-responder for that time point only.

In an effort to find evidence-based techniques to improve retention and recruitment to RCTs, a study within a trial (SWAT) will be undertaken. This will examine whether there is a difference in questionnaire response rate by printing the UCLA Activity Scale on pink rather than white coloured paper at the 6 month time point. Results of this SWAT will be reported separately to the 'host' trial.

## Outcome measures
The data collection schedule is presented in table 2.

### Primary outcome
► UCLA Activity Scale:[25] This is a reliable and valid self-reported tool to assess physical activity.[33 34] It assesses global activity levels with a grading system of 1 out of 10 points where one equates to 'wholly inactive, dependent on others and cannot leave residence' and 10 refers to 'regularly participants in impact sports'.[25]

### Secondary outcomes
► LEFS:[26] This is a valid and reliable measure of functional impairment in patients with lower extremity musculoskeletal conditions (Mehta *et al*, 2016).[35]
► Oxford Hip Score[27] or Oxford Knee Score:[28] Both measures have demonstrated good validity and reliability for people undergoing THR or TKR.[36]
► Numerical Rating Scale (NRS) for Pain: The NRS has been previously reported as reliable and valid to assess pain.[37]
► Generalised Self-Efficacy Scale:[29] This is a reliable and valid measure of self-efficacy.[38 39]
► The Tampa Scale for Kinesiophobia:[30] This is a valid and reliable measure of kinesiophobia.[40 41]
► HADS:[31] This has been shown as a reliable and valid measure of anxiety and depression.[42]
► Complications or adverse events (self-reported) which may include: wound or joint infection, joint dislocation, delayed hospital discharge (measured by length of stay), falls or musculoskeletal injuries and exacerbations of multi-site joint pain. Adverse events and serious adverse events will be reported as per the clinical trial unit's SOP.
► Health economic and health resource utilisation: We will collect HRQoL using the EQ-5D-5L.[33] Primary sources (ie, participant log books) will be used to record the duration of the assessment visits, number and duration of scheduled group sessions, equipment, consumables, educational material, behaviour change intervention training time; number and duration of scheduled telephone calls. Frequency of healthcare resource use will be collected through patient self-reported questionnaires at follow-ups. The categories of NHS resource use that will be collected include treatment costs, outpatient visits, any related-inpatient admissions, rehabilitation visits and any treatment for recurrences and adverse events. Non-medical costs such as care giving and productivity loss for those in employment at each follow-up will be collected.

**Table 2** Data collection schedule

| Data | Time points and mode of data collection | | | |
|---|---|---|---|---|
| | Baseline | Intervention period | Month 6 post-randomisation | Month 12 post-randomisation |
| | Face-to-face | Face-to-face | Postal | Postal |
| Age (years) | ■ | | | |
| Gender | ■ | | | |
| Weight (kg) | ■ | | | |
| Height (cm) | ■ | | | |
| Admission date | ■ | | | |
| Operative procedure (THR or TKR) | ■ | | | |
| Site of joint replacement | ■ | | | |
| Duration of hip or knee symptoms | ■ | | | |
| Presence and location of multi-joint pain | | | ■ | ■ |
| ASA grade | ■ | | | |
| AMTS | ■ | | | |
| List of medical comorbidities | ■ | | | |
| Physiotherapist exercise class log | | ■ | | |
| Patient in-session and home exercise diary | | ■ | | |
| Behaviour change intervention log (group and telephone) (physiotherapist completed) | | ■ | | |
| Charlson Comorbidity Score | ■ | | | |
| Employment status and current occupation (when appropriate) | ■ | | ■ | ■ |
| UCLA Activity Score | ■ | | ■ | ■ |
| Lower Extremity Functional Scale | ■ | | ■ | ■ |
| Oxford Hip or Knee Score | ■ | | ■ | ■ |
| Numerical Rating Scale – Pain | ■ | | ■ | ■ |
| Generalised Self-Efficacy Scale | ■ | | ■ | ■ |
| Tampa Scale for Kinesiophobia | ■ | | ■ | ■ |
| Hospital Anxiety and Depression Scale | ■ | | ■ | ■ |
| EQ-5D-5L | ■ | | ■ | ■ |
| Health economic/health utilisation questionnaire | ■ | | ■ | ■ |
| Complications and adverse events | | | ■ | ■ |

Shaded areas represent where these data are being collected.
Assessment intervals following randomisation; each follow-up interval ±1 month.
AMTS, Abbreviated Mental Test Score; ASA, American Society of Anesthesiologists classification; cm, centimetres; kg, kilograms; THR, total hip replacement; TKR, total knee replacement; UCLA, University of California, Los Angeles.

## Data analysis

### Sample size

Originally, 250 participants (125 per arm) are required to detect a standardised effect size of 0.4 with 80% power and 5% (two-sided) significance, and allowing for 20% loss to follow-up. These calculations are based on the primary outcome, UCLA Activity Scale, at 12 months, assuming a baseline SD of 2.5 and a between-group difference of 1.[33] The minimally clinically important difference has been reported as a within person difference of 0.92 points.[33]

The sample size was increased to 260 to account for the amendment in randomisation procedure from 1:1 to 2:1 group allocation, as described earlier.

## Statistical analysis

A detailed statistical analysis plan (SAP) will be drafted early in the trial and will be finalised prior to any primary outcome analysis. This will be reviewed and will receive input from the Trial Steering Committee (TSC) and the DSMC.

All analyses will be undertaken on the intent-to-treat population, that is, patients will be analysed as they were randomised regardless of the treatment received. Sensitivity analyses will be undertaken on the per-protocol population to assess a range of potential biasses that could have resulted from loss to follow-up, protocol deviations and withdrawal (including mortality). Numerical

and graphical summaries of all data will be presented including descriptions of missing data at each level.

Estimates of treatment effects will be reported with 95% CIs. The primary outcome measure, UCLA Activity Scale at 12 months post-randomisation, will be analysed using a mixed effects linear regression model adjusted for baseline activity, 6 month time point and the stratification factors. Centre will be included as a random-effect to take account of their potential heterogeneity and type of operation, comorbidity index and other time points (baseline and 6 months) will be included as fixed-effects. Treatment by 6 month time point interaction will also be included in the model to allow time specific treatment effects to be calculated The adjusted difference in the means with corresponding 95% CI will be reported together with the mean and 95% CIs for each treatment group. If not normally distributed, then transformation to normality will be considered. If this is not possible, non-parametric techniques (for example Mann-Whitney or Kruskal-Wallis test) will be used with no adjustment. Consistency of any observed intervention effect will be explored using forest plots for various subgroups including site, type of operation (TKR and THR) and comorbidity index using interactions. No formal testing of interaction effects will be undertaken as the trial is not powered to detect these and the effect of the intervention is expected to be similar in the THR and TKR subgroups, although their baseline physical activity may be different, which will be taken into account in the analysis.

Similar methods will be used to analyse continuous secondary outcomes. Binary and categorical secondary outcomes, such as complications, will be tabulated to show frequencies and percentages in each arm. $\chi^2$ tests or logistic regression will be used to assess statistical significance.

Missing data will be minimised by careful data management. Missing data will be described with reasons given where available. The number and percentage of individuals in the missing category will be presented by treatment arm.

The nature and mechanism for missing variables and outcomes will be investigated, and if appropriate, multiple imputation will be used. Sensitivity analyses will be undertaken assessing the underlying missing data assumptions. Any imputation techniques will be fully described in the SAP. Subgroup analyses, complies average causal effect analysis and mediation analyses will be undertaken after being fully specified in the SAP. Compliance with the intervention is defined as attending at least four of six group sessions with a minimum of three participants in the group for these sessions and receiving at least two of the three follow-up phone calls. A priori mediation analysis moderators will include self-efficacy, fear avoidance, psychological distress to compare the mediation pathways presented in the BeST (BackSkills Training) intervention[43] to the PEP-TALK intervention.

All data collected on data collection forms will be used, since only essential data items will be collected. No data will be considered spurious in the analysis since all data will be checked and cleaned before analysis.

## Health economics
Respecting an *efficient trial design*, only if the data indicates clinical effectiveness of the experimental intervention (for the UCLA Activity Scale and LEFS), will additional funding be sought to analyse the health economic and utilisation data, to determine the cost-effectiveness of the intervention.

## Data management
All data will be processed according to the Data Protection Act 2018.[44] All documents will be stored safely in confidential conditions. Trial-specific documents, except for the signed consent form and follow-up contact details, will refer to the participant with a unique study participant number, not by name. Participant identifiable data will be stored separately from trial data. All trial data will be stored securely in offices or online in secure trial databases, only accessible by the central trial team in Oxford and authorised personnel.

## Trial status
The trial is funded for 36 months and commenced in August 2018. Recruitment is expected to be complete by April 2020 with the final follow-up visit for the final participant completed by April 2021. The trial will be completed by August 2021.

## Site locations
Orthopaedic services providing THR and TKRs in nine hospital trusts (eight recruiting sites) and health providers: (Norfolk and Norwich University Hospitals NHS Foundation Trust, Lewisham and Greenwich University Hospitals NHS Foundation Trust, Oxford University Hospital NHS Foundation Trust, City Hospitals Sunderland NHS Foundation Trust, Norwich Spire, Barts Health NHS Trust, North Middlesex University Hospital NHS Trust, St George's University Hospital NHS Foundation Trust (treatment site only), Epsom and St Helier University Hospitals NHS Foundation Trust (recruitment site only)).

## Patient and public involvement
Patient involvement began during protocol development and continues throughout the trial. A patient-member will attend all TSC meetings. The same patient-member is a co-investigator, providing insights into the trial conduct, particularly on data collection processes, and will help interpret the findings to inform on the implications of the research during the trial's dissemination phase.

## ETHICS AND DISSEMINATION
Ethical approval was gained from the South Central (Oxford B) Research Ethics Committee (Approval Date: 23 October 2018; Reference Number: 18/SC/0423). The trial was prospectively registered. A DSMC and TSC was

appointed to independently review the data on safety, protocol adherence and recruitment to the trial. Direct access will be granted to authorised representatives from the sponsor and host institution for monitoring and/or audit of the trial to ensure compliance with regulations. Anonymised data will be shared outside the research team when required. Researchers outside the trial team may formally request for a specific data set using a data request form, which will be part of the Data Management Plan. All such requests will need to be approved by the TMG.

Reporting of the trial will be consistent with the CONSORT 2010 Statement (patient reported outcomes and non-pharmcological interventions)[45] and Template for Intervention Description and Replication (TIDieR)[46] guidelines. A summary of the results and trial materials will be made available via the trial website on completion of the trial. We will submit the final report to a peer-reviewed academic journal.

## DISCUSSION

This paper presents the research protocol for the PEP-TALK trial. Following a THR and TKR, only 50% of people reach WHO recommended levels of physical activity.[15 47 48] Those who are least likely to meet these levels are people with a higher body mass index and with comorbidities.[49] These patients have the most to gain from being more physically active. If this trial's experimental intervention is shown to be effective, this intervention could have a significant and sustained impact on improving the management of comorbidities such as diabetes, cardiovascular diseases, depression and hypertension for over 103 000 people annually in England and Wales.[1] It is proposed that, in such an instance, this should be considered for implementation in healthcare services. This could help address the global challenge which multi-morbidities and an ageing population are expected to have on health and social care services.

**Author affiliations**
[1]Nuffield Department of Orthopaedics, Rheumatology and Musculoskeletal Sciences, University of Oxford, Oxford, UK
[2]Faculty of Medicine and Health Sciences, University of East Anglia, Norwich, UK
[3]Oxford Clinical Trials Unit, University of Oxford, Oxford, UK
[4]CSM, University of Oxford, Oxford, UK
[5]University of London St George's Molecular and Clinical Sciences Research Institute, London, UK
[6]College of Medicine and Health Sciences, University of Exeter, Exeter, UK

**Collaborators** The PEP-TALK Trial Collaborators: Mr Steve Algar (Banbury, PPI Representative), Dr Zara Hansen (ZH) (University of Oxford), Professor Karen Barker (Principal Investigator - Oxford University Hospital NHS Foundation Trust (OUH), Mr Ian Smith (Principal Investigator - Lewisham and Greenwich University Hospitals NHS Foundation Trust), Professor Iain McNamara (Principal Investigator (Norfolk and Norwich University Hospitals NHS Foundation Trust (NNUH) & Spire Norwich), Mr Michael Dunn (Principal Investigator - St George's University Hospital NHS Foundation Trust), Mrs Dawn Lockey (Principal Investigator - City Hospitals Sunderland NHS Foundation Trust), Mr Sonny Driver (Principal Investigator – North Middlesex University Hospital NHS Trust), Mrs Jamila Kassam (Barts Health NHS Trust), Mr Peter Penny (Norwich Spire, Norwich), Mrs Celia Woodhouse, Mrs Tracey Potter and Mrs Helena Daniell (NNUH), Mr Alex Herring and Mrs Yan Cunningham (City Hospitals Sunderland NHS Foundation Trust), Irrum Afzal (South West London Elective Orthopaedic Centre), Mr Maninderpal Matharu (Barts Health NHS Trust) and Mrs Tamsin Hughes, Ms Erin Hannink and Mrs Michelle Moynihan (OUH). Oversight Committee Membership: TSC Members: Professor David Deehan (Newcastle University), Dr Emma Godfrey (Kings College London), Dr Neil Artz (University of the West of England), Mr Steve Algar (PPI representative). DSMC Members: Dr Lindsey Smith (University of the West of England & Weston Area Health NHS Trust), Dr Dipesh Mistry (University of Warwick) and Mr Paul Baker (South Tees Hospital NHS Foundation Trust).

**Contributors** TS, BF, SD, SL, AO, CH, VB and SP researched the topic and devised the study. TS, SP, AO, BF, SD, CH, VB, MEP and SL provided the first draft of the manuscript. AO and SD provided statistical oversight. TS, SP, BF, AO, SD, CH, VB, MEP and SL contributed equally to manuscript preparation. TS acts as a guarantor.

**Funding** The research is supported by the National Institute for Health Research (NIHR) Research for Patient Benefit grant (PB-PG-1216-20008). Trial Sponsor: University of Oxford, Clinical Trials and Research Governance Team, Joint Research Office, 1st floor, Boundary Brook House, Churchill Drive, Headington, Oxford OX3 7GB; email: ctrg@admin.ox.ac.uk.

**Disclaimer** TS is supported by the National Institute for Health Research (NIHR) Oxford Biomedical Research Centre (BRC). The views expressed are those of the author(s) and not necessarily those of the NHS, the NIHR or the Department of Health and Social Care.

**Competing interests** None declared.

**Patient and public involvement** Patients and/or the public were involved in the design, or conduct, or reporting, or dissemination plans of this research. Refer to the Methods section for further details.

**Patient consent for publication** Not required.

**Provenance and peer review** Not commissioned; externally peer reviewed.

**ORCID iD**
Toby O Smith http://orcid.org/0000-0003-1673-2954

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
