## [Reviewer comments · BMJ Open]

ARTICLE DETAILS

TITLE (PROVISIONAL)	Behaviour change physiotherapy intervention to increase physical activity following hip and knee replacement (PEP-TALK): study protocol for a pragmatic randomised controlled trial
AUTHORS	Smith, Toby O.; Parsons, Scott; Fordham, Beth; Ooms, Alexander; Dutton, Susan; Hing, Caroline; Barber, Vicki; Png, May ee; Lamb, Sarah

VERSION 1 - REVIEW

REVIEWER	AF Lenssen Maastricht University Medical Center, Maastricht Netherlands
REVIEW RETURNED	04-Feb-2020

GENERAL COMMENTS	In my opinion a well written study protocol for an interesting study. I've got one major and a few minor points of concern. Major point of concern is the lack of insight in the responsiveness of the main outcome assessment. The UCLA score is a 10 point outcome score which does not incorporate frequency and intensity of physical activity participation and therefore a stable score may not be related to unchanged physical activity participation. Frequency and intensity may well differ after surgery, without changes in UCLA score. -page 4; Concerning the study rationale , Authors do not state whether or not patients complain about their level of physical functioning and how that level relates to global recommendations on physical activity for Health. I would suggest adding these point in the introduction to increase the awareness that limited physical activity is a topic that needs to be addressed. -Page 6 lines 29-32; I understand that blinding patients and caregivers is not possible, but why weren't outcome assessors blinded? -page 10 lines 43-44 ; Authors describe the minimal important difference, are data available concerning the minimal detectable difference as well?
---

REVIEWER	Enrica Papi Imperial College London
REVIEW RETURNED	06-Feb-2020

GENERAL COMMENTS	This is a very interesting paper and easy to read, certain parts may benefit for more explanation to add clarity to the manuscript. Minor
---

	suggestions are detailed below: Background: I suggest the aim to be re-written following the PICOT format. Particularly as it is now it is not clear at which time point the study primary outcome is evaluated. Methods and design: Trial Design: in the abstract it is also mentioned the study is multi-centre. This should also be added in here and the number of the centre involved should be added to together to where they are (e.g. UK vs EU?!) Participants: which is the sample size? I noticed you have the sample size in the data analysis section but I think it is clearer for the reader if it is mentioned when the recruitment criteria are described. This also because there is a reference to it on page 6 line 43. Randomisation, Blinding...: A very brief description of what control and intervention groups are should be added. Experimental intervention: Overall I think this is the part that is less clear. There is reference to post-group telephone-based sessions in the first paragraph, what are these and what for? I realised as I was reading that these are described later on but referenced to as 'telephone follow-up calls'. I suggest to be consistent with the name and in the first paragraph add a brief explanation of the aim of the calls. Page 7 line 28: references needed for 'evidence-based behaviour change techniques'. Who delivers these? What is the expertise of those who deliver them? 'home-practice element': it is not clear what this comprises and how it is delivered, what are you expecting participants to do? Is this only over the 6-week intervention period? Please clarify Co-intervention: how do you account for additional treatment effects? Page 9 line 26: 'primary outcome' just say what that is Page 9 Line 30-35: Not sure this is relevant and add anything to the current manuscript. I would suggest to remove it Table 2 is not very clear, what is the difference between grey and white areas? Specify When is the primary outcome evaluated? At 6 week? The time line of outcomes evaluation is missing, do you compare outcomes at different time points?
--	---

VERSION 1 – AUTHOR RESPONSE

Response to Reviewer's Comments

Reviewer 1

Comment: In my opinion a well written study protocol for an interesting study. I've got one major and a few minor points of concern.

Response: Thank you for your comments. We have responded to these below.

Comment: Major point of concern is the lack of insight in the responsiveness of the main outcome assessment. The UCLA score is a 10 point outcome score which does not incorporate frequency and intensity of physical activity participation and therefore a stable score may not be related to unchanged physical activity participation. Frequency and intensity may well differ after surgery, without changes in UCLA score.

Response: Thank you for this comment. The choice for primary outcome measure was made after very careful assessment of the literature. The evidence-base surrounding the UCLA physical activity score to assess physical activity for people following joint replacement surgery is compelling both from a reliability and validity perspective (Terwee et al, 2011). Whilst the instrument provides a clear insight into physical activity performance in a patient-reported outcome measure (PROM), it was deemed most appropriate to assess between-group differences for our trial. There is also evidence that the UCLA activity score can discriminate between insufficiently active and sufficiently active patients undergoing both total knee and total hip replacements (Florian et al, 2009). Whilst we considered the use of accelerometry and other approaches to physical activity data collection, the attraction of all our outcomes being collected as PROMS and not performance-based, was felt as most appropriate. Finally, given that we have commenced the trial, and powered the design on the primary outcome, it would be inappropriate to now change this measure. Based on this, we have not amended the primary outcome.

Comment: Page 4; Concerning the study rationale , Authors do not state whether or not patients complain about their level of physical functioning and how that level relates to global recommendations on physical activity for Health. I would suggest adding these point in the introduction to increase the awareness that limited physical activity is a topic that needs to be addressed.

Response: We have provided supporting text as suggested to strengthen the rationale (Background, Paragraph 3, Lines 5-8).

Comment: Page 6 lines 29-32; I understand that blinding patients and caregivers is not possible, but why weren't outcome assessors blinded?

Response: All outcomes were patient-reported outcome measures and therefore no outcome assessors were used in this trial design. Accordingly outcome assessor blinding was not required.

Comment: Page 10 lines 43-44 ; Authors describe the minimal important difference, are data available concerning the minimal detectable difference as well?

Response: There is no mention of a formal minimal detectable difference for the UCLA Score in the current literature. However, given the existing evidence, the UCLA score is sensitive enough to detect a meaningful between-group difference for a study of this size (SooHoo et al, 2015).

Reviewer 2

Comment: This is a very interesting paper and easy to read, certain parts may benefit for more explanation to add clarity to the manuscript. Minor suggestions are detailed below:

Response: Thank you for your comments. We have responded to these below.

Comment: Background: I suggest the aim to be re-written following the PICOT format. Particularly as it is now it is not clear at which time point the study primary outcome is evaluated.

Response: We have revised the aim of the trial to be more explicitly framed around the PICOT framework (Background, Paragraph 5, Lines 10-13).

Comment: Methods and design: Trial Design: in the abstract it is also mentioned the study is multi-centre. This should also be added in here and the number of the centre involved should be added to together to where they are (e.g. UK vs EU?!)

Response: We have included this as suggested (Methods and Design, Trial Design, Lines 1 & 3-4).

Comment: Participants: which is the sample size? I noticed you have the sample size in the data analysis section but I think it is clearer for the reader if it is mentioned when the recruitment criteria are described. This also because there is a reference to it on page 6 line 43.

Response: We have provided brief details here to provide context as suggested (Methods and Design, Participants, Line 1) but further information on sample size is provided as per the original paper

Comment: Randomisation, Blinding...: A very brief description of what control and intervention groups are should be added.

Response: We have incorporated this as suggested (Methods and Design, Randomisation, Blinding and Allocation Concealment, Paragraph 1, Lines 1-2).

Comment: Experimental intervention: Overall I think this is the part that is less clear. There is reference to post-group telephone-based sessions in the first paragraph, what are these and what for? I realised as I was reading that these are described later on but referenced to as 'telephone follow-up calls'. I suggest to be consistent with the name and in the first paragraph add a brief explanation of the aim of the calls.

Response: As suggested, we have revised the text to be more consistent in the labelling of the telephone call. We have also simplified the text so it is clearer as to what these were, bridging to a more comprehensive description in Paragraph 4 of this section (Methods and Design, Intervention, Experimental Intervention, Paragraph 1, Lines 5-6).

Comment: Page 7 line 28: references needed for 'evidence-based behaviour change techniques'. Who delivers these? What is the expertise of those who deliver them? 'home-practice element': it is not clear what this comprises and how it is delivered, what are you expecting participants to do? Is this only over the 6-week intervention period? Please clarify

Response: As suggested, we have provided further information regarding the intervention and the training provided to ensure that the intervention could be delivered as per protocol (Intervention, Experimental Intervention, Paragraph 3, Lines 4-11). As suggested by the reviewer, we have also provided further information regarding the home-practice element (Intervention, Experimental Intervention, Paragraph 5, Lines 3-11).

Comment: Co-intervention: how do you account for additional treatment effects?

Response: We will be monitoring for co-interventions. This is a pragmatic randomised controlled trial. Accordingly, we will not control for these but will monitor for these within the health utilisation questionnaire to assess for between-group differences.

Comment: Page 9 line 26: 'primary outcome' just say what that is

Response: This has been corrected as suggested (Methods and Design, Assessments, Baseline Assessments, Paragraph 3, Line 7; Paragraph 4, Line 3).

Comment: Page 9 Line 30-35: Not sure this is relevant and add anything to the current manuscript. I would suggest to remove it

Response: We believe that this refers to the SWAT. This is an embedded study within the trial and therefore we feel that this has merit in the reporting of this trial within the protocol for completeness. We have therefore elected to keep these 4 lines in the text. If the editorial team and reviewers feel strongly against this, we would be happy to reconsider this decision.

Comment: Table 2 is not very clear, what is the difference between grey and white areas? Specify

Response: We have specified in the text that the shaded areas represent when data are collected (Table 2, Footnote).

Comment: When is the primary outcome evaluated? At 6 week? The time line of outcomes evaluation is missing, do you compare outcomes at different time points?

Response: In the Data Analysis section, we have described the analyses to be undertaken. We have stated the primary end-point being 12 months. The other time points (baseline and 6 months) will be included in the main analysis as fixed effects in a mixed effects model. Following the reviewer's comments, we have made this more explicit in the revised paper (Data Analysis, Statistical Analysis, Paragraph 3, Lines 3 and 5-7).

VERSION 2 – REVIEW

REVIEWER	AF Lenssen Maastricht University Medical Center, Maastricht Netherlands
REVIEW RETURNED	08-Apr-2020

GENERAL COMMENTS	I think the authors wrote a clear cut protocol, which in my opinion is acceptable for publication.
--

REVIEWER	enrica papi Imperial College London, UK
REVIEW RETURNED	30-Mar-2020

GENERAL COMMENTS	thank you for responding to my queries. no more comments to add.
--